# Sensitive and Facile HCOOH Fluorescence Sensor Based on Highly Active Ir Complexes’ Catalytic Transfer Hydrogen Reaction

**DOI:** 10.3390/molecules27217431

**Published:** 2022-11-01

**Authors:** Caimei Zhang, Wenjuan Zhang, Yiran Wu, Bo Peng, Chunyuan Tian, Feng Luan, Wen Sun, Xuming Zhuang, Lijun Zhao

**Affiliations:** 1School of Chemistry & Chemical Engineering, Yantai University, Yantai 264005, China; 2State Key Laboratory of Fine Chemicals, Dalian University of Technology, Dalian 116024, China

**Keywords:** HCOOH detection, Ir complexes, catalytic hydrogenation, fluorescence probe

## Abstract

With several major polarity and weak optical properties, the sensitive detection of HCOOH remains a major challenge. Given the special role of HCOOH in assisting in the catalytic hydrogenation process of Ir complexes, HCOOH (as a hydrogen source) could rapidly activate Ir complexes as catalysts and further reduce the substrates. This work developed a facile and sensitive HCOOH fluorescence sensor utilizing an optimal catalytic fluorescence generation system, which consists of the phenyl-pyrazole-type Ir-complex PP-Ir-Cl and the coumarin-type fluorescence probe P-coumarin. The sensor demonstrates excellent sensitivity and specificity for HCOOH and formates; the limits of detection for HCOOH, HCOONa, and HCOOEt_3_N were tested to be 50.6 ppb, 68.0 ppb, and 146.0 ppb, respectively. Compared to previous methods, the proposed sensor exhibits good detection accuracy and excellent sensitivity. Therefore, the proposed HCOOH sensor could be used as a new detection method for HCOOH and could provide a new design path for other sensors.

## 1. Introduction

With global warming, carbon dioxide (CO_2_) fixation and utilization have become global hot research topics [1,2,3]. Catalytic technologies (including chemical catalytic technology, electrocatalytic technology, and photocatalytic technology) have attracted more and more attention since they could successfully convert carbon dioxide to formic acid (HCOOH), methanol (MeOH), and other chemicals [4,5,6,7]. Among these manufactured chemicals, HCOOH and formate have been widely used in oil exploitation; leather, textile, and oxalic acid production; and other fields [8,9,10,11,12,13]. In addition, HCOOH is an important liquid hydrogen storage material [14,15,16,17], which could be used as a potential feedstock for fuel cells. In the bioanalysis field, HCOOH in urine is also used as an important biomarker of human exposure to formaldehyde [18]. Based on these important performances, the analysis of HCOOH has attracted much attention. However, the sensitive detection of HCOOH still faces many challenges due to its severely high polarity, strong acidity, and weak optical property, which must be aided by the expensive high-performance liquid chromatography (HPLC), nuclear magnetic resonance (NMR) techniques, and some titrimetric analyses.

Currently, the main industrial analytical method of HCOOH is the potassium permanganate (KMnO_4_) oxidation indirect iodine quantity method, which can only analyze high concentrations of HCOOH [19]. This method mainly benefits from the good reducibility of HCOOH. HCOOH could be rapidly oxidized into carbonic acid by KMnO_4_, and the excess KMnO_4_ quantitatively oxidizes potassium iodide (KI) into iodine (I_2_). The concentration of HCOOH could be obtained by titrating the generated I_2_ with sodium thiosulfate (Na_2_S_2_O_3_). The analysis of trace amounts of HCOOH is currently performed using expensive instruments, such as HPLC [20]. The detection limits can usually be reached between 0.1 mg/mL and 1.0 mg/mL. Even so, when HCOOH is detected using HPLC, specialized organic acid analysis columns (such as Aminex HPX-87H Ion Exclusion Column) are necessary due to the severely high polarity. Thus, there is an urgent need to develop more sensitive and convenient HCOOH detection methods.

In various catalytic hydrogenation processes, HCOOH/HCOONa is often used as an important hydrogen source, which has many advantages, such as low cost, nontoxicity, stability, and good water solubility [21,22]. In particular, some metal iridium complexes have shown excellent catalytic hydrogenation performance for various important oxidation molecules (such as quinone derivatives and nicotinamide adenine dinucleotide) in the presence of HCOOH [23]. In previous work, we have successfully developed an ultra-highly active [N^N¯] iridium complex [Cp*Ir(pba)Cl] (Cp* = pentamethylcyclopentadiene, pba = 4-(picolinamido)benzoic acid), which exhibits the highest hydrogenation activity for NAD^+^ in the physiological condition (turnover frequency (TOF) is 7825 h^−1^ at 37 °C) when HCOONa is used as a hydrogen source [24]. Based on the excellent hydrogen donor capacity of HCOOH and the development of highly active Ir-complex catalysis, we aimed to build a sensitive and convenient HCOOH fluorescence sensor based on highly active Ir-complexes’ catalytic transfer hydrogen reactions.

Here, a novel HCOOH fluorescence sensor was developed based on highly active catalytic hydrogenation reactions, which consists of highly active and stable phenyl pyrazole Ir-complex catalysis and a sensitive quinone hydrogenation fluorescence turn-on probe. For a complete catalytic hydrogenation system, the hydrogen sources (such as HCOOH and H_2_) are also essential, besides the catalysts and substrates. As shown in Figure 1, in the absence of HCOOH, the substrates of the quinone fluorescence probe cannot be reduced by an Ir complex and hardly any fluorescence signals are detected due to the intramolecular charge transfer (ICT) effect. However, in the presence of HCOOH, the quinone in the probe molecules could be quickly reduced to hydroquinone and removed from the probe structure. Thus, the hydroxyl group of coumarin fluorochrome is free again, which will produce a strong fluorescence signal. The optimal hydrogenation fluorescence response system was obtained by studying the interaction between the catalysts and the probes. A super-active catalyst could easily lead to fluorescence decrease in fluorochrome due to the over-reduction effect. However, a less active catalyst may struggle to fully reduce the quinone of the probe molecules to hydroquinone, and the obtained fluorescence signals were also very weak. Furthermore, the optimal hydrogenation fluorescence response system demonstrated excellent sensitivity and selectivity for HCOOH and formates (HCOONa and HCOOEt_3_N). Compared to NMR analytical methods, the proposed sensor exhibits good detection accuracy and higher sensitivity, which could be used as a potential sensitive detection method for trace HCOOH analysis and could provide a new design path for more sensors.

## 2. Results

### 2.1. Detection Principle of the Ir-Complex Catalytic Fluorescence Sensor for HCOO^−^

As an important hydrogen source, HCOOH has been widely applied in various hydrogenation catalytic reactions. In previous work, we have systematically studied the fluorescence response performance of several main iridium complexes for redox-type fluorescence probes in the presence of HCOONa. The previous studies’ results demonstrated that C^N-type phenyl pyrazole Ir complex PP-Ir-Cl could best catalyze the fluorescence probe P-coumarin to convert it into coumarin dye, which leads to a higher fluorescence signal and a better formic acid detection performance. Therefore, in order to establish the optimal HCOOH-sensor-based catalytic hydrogenation reaction, PP-Ir-Cl and P-coumarin were selected as the hydrogenation catalyst and the fluorescence probe, respectively.

### 2.2. Optimization of Experimental Parameters

In order to obtain the optimal conditions, we investigated the effects of various experimental parameters, including the pH of the reaction solution, the reaction solvents, temperature, the amount of the catalyst PP-Ir-Cl, and the reaction time, on the fluorescence response performance of the HCOOH sensor. As shown in Figure 1A, the fluorescence intensity of the probe increases with an increase in pH from 2.0 to 8.0, regardless of whether HCOONa is present or not, which could be attributed to the coefficient of the unstable fluorescence probe P-coumarin at a higher pH condition and the self-fluorescence pH response performance of the probe substrate 7-hydroxycoumarin. In order to obtain an optimal detection performance, the signal-to-noise ratios (SNR) of different pH conditions were compared. The optimal SNR was found to be 34 at pH = 4.0, which means 4.0 was the optimal pH for the probe P-coumarin. Furthermore, the effects of different organic solvents on the fluorescence response of the HCOOH sensor were studied under the optimal pH condition (Figure 1B). The results showed that protic solvents, such as methanol and ethanol, could ensure both a high fluorescence signal with HCOONa and a low background noise without HCOONa. Therefore, 10 *v*/*v*% of methanol was used in subsequent experiments to obtain a high fluorescence signal and a stable SNR. In a catalytic hydrogenation reaction, the temperature usually has a significant influence on the reaction efficiency. Thus, the reaction temperature of the HCOOH sensor was also investigated in the range from 20 to 80 °C under a pH = 4.0 condition and in the presence of 1 mM of HCOONa and 10 *v*/*v*% of methanol (Figure 1C). We found that the fluorescence intensity of the sensor increased with an increase in temperature when it was below 60 °C due to the influence of reaction thermodynamics, and slightly decreased at 80 °C, which might be attributed to the over-reduction of the catalyst PP-Ir-Cl for the probe substrate 7-hydroxycoumarin. A temperature of 60 °C was demonstrated to possess the highest fluorescence signal and, thus, could be used as the optimal reaction temperature in subsequent experiments. In order to obtain a better detection performance, the effects of three different amounts of the catalyst PP-Ir-Cl (1.0 µM, 5.0 µM, and 10.0 µM) on the fluorescence signals of the sensor were also investigated in an HAc-NaAc buffer (0.2 M, pH = 4.0) containing 1.0 mM of HCOONa, 100 µM of P-coumarin, and 10 *v*/*v*% of methanol at 60 °C for 30 min (Figure 1D). The best fluorescence signal and SNR (34) were obtained at 5.0 µM of the catalyst amount, which could be attributed to the appropriate reduction degree of the 5.0 µM of PP-Ir-Cl for the P-coumarin. Under the abovementioned optimal conditions, we further studied the variation in the fluorescence intensity of the sensor with reaction time in the presence of 1.0 mM of HCOONa (Figure 1E). The fluorescence intensity of the sensor increased with an increase in reaction time from 0 to 30 min and reached the highest value at 30 min. Finally, in order to evaluate the fluorescence stability of the hydrogenation product 7-hydroxycoumarin under the optimal reaction conditions, three different concentrations of 7-hydroxycoumarin (20 µM, 50 µM, and 100 µM) in an HAc-NaAc buffer (0.2 M, pH = 4.0) containing 1.0 mM of HCOONa, 10 *v*/*v*% of methanol, and three different concentrations of PP-Ir-Cl (0 µM, 1.0 µM, and 10 µM) were incubated at 60 °C for 30 min, and then, the fluorescence intensity was tested. As shown in Figure 1F, the fluorescence intensity of the dye 7-hydroxycoumarin is almost unaffected by the catalyst and the reaction environment, which shows the extreme stability of 7-hydroxycoumarin in the proposed sensor environment.

### 2.3. Ir-Complex Catalytic Fluorescence Sensor for the Detection of HCOOH and Formates (HCOONa and HCOOEt_3_N)

Under the optimal experimental conditions, the analytical performance of the developed Ir-complex catalytic fluorescence sensor was first investigated by detecting various concentrations of HCOOH. As shown in Figure 2A, the fluorescence intensity increases with an increase in HCOOH concentration in the range from 3.9 µM to 2.0 mM, indicating that more HCOOH could provide more protons and could catalyze the probe to produce more fluorescence substance 7-hydroxycoumain. The recovery values of the fluorescence intensity are in a linear relationship with the logarithm of the HCOOH concentration in the range from 7.8 µM to 0.5 mM, and the limit of detection (LOD) was calculated to be 1.1 µM (50.6 ppb) according to the 3σ/s. Figure 2B demonstrates that the linear equation is I = 410.6logC_HCOOH_ − 136.53 with a linear correlation coefficient of 0.9871, where I is the recovery intensity value of fluorescence. Compared to other proposed methods of HCOOH detection, the HCOOH fluorescence sensor exhibits a lower LOD and a satisfactory linear range (Table 1).

Moreover, the Ir-complex catalytic fluorescence sensor was used to detect two important formates: HCOONa and HCOOEt_3_N. As shown in Figure 3A, the fluorescence intensity increases with an increase in HCOONa concentration in the range from 0.1 µM to 1.0 mM, which indicates that more HCOONa could help catalyze the probe to produce more of the fluorescence substance 7-hydroxycoumain. The recovery values of the fluorescence intensity are in a linear relationship with the logarithm of the HCOONa concentration in the range from 0.1 µM to 50 µM, and the limit of detection (LOD) was calculated to be 1.0 µM (68 ppb) according to the 3σ/s. Figure 3B demonstrates that the linear equation is I = 667.1logC_HCOONa_ − 1392.5 with a linear correlation coefficient of 0.9980, where I is the recovery intensity value of fluorescence. Figure 3C shows that the fluorescence intensity increases with an increase in HCOOEt_3_N concentration in the range from 0.5 µM to 1.25 mM, which indicates that more HCOOEt_3_N could also help catalyze the probe to produce more fluorescence substance 7-hydroxycoumain. The recovery values of the fluorescence intensity are in a linear relationship with the logarithm of the HCOOEt_3_N concentration in the range from 10 µM to 500 µM, and the limit of detection (LOD) was calculated to be 1.0 µM (146 ppb) according to the 3σ/s. Figure 3D demonstrates that the linear equation is I = 1950.5logC_HCOOEt3N_ − 1449.2 with a linear correlation coefficient of 0.9902, where I is the recovery intensity value of fluorescence.

### 2.4. The Selectivity and Repeatability of the Ir-Complex Catalytic Fluorescence Sensor

As an important organic acid, the detection of HCOOH is easily disturbed by other organic acids, especially ethanoic acid and propanoic acid, due to similar physical and chemical properties. Thus, the specificity of the HCOOH sensor was also evaluated using various interfering substances (5.0 mM), instead of HCOOH or formate (1.0 mM), under the same experimental conditions. Various inorganic and organic salts, and organic acids, including methanol MeOH, ethanol EtOH, tert-Butanol, NADH, oxalic acid, sodium sulfate, 2-bromopropionic acid, 4-bromobutyric acid, mandelic acid, NH_2_(CH_2_)_5_COOH, glucose, KCl, acetic acid HAc, NaAc, NaH_2_PO_4_, Na_2_HPO_4_, sodium citrate, and NaCl, were used as the interfering agents because they are common in the environment. As shown in Figure 4A, compared to HCOOH and formate, the fluorescence intensity of the interfering agents is negligible except for NADH and sodium citrate. The stronger fluorescence intensity from NADH and sodium citrate might be attributed to their stronger reducibility. These results indicate that the developed HCOOH sensor has a strong anti-interference ability and can accurately detect HCOOH in complex samples. For a sensitive sensor, its repeatability is also very important. In order to investigate the repeatability of the sensor, we carried out five repeated experiments under the same experimental conditions. As shown in Figure 4B, the fluorescence signals of the five repeated experiments are almost consistent, which demonstrate that the established HCOOH sensor is stable.

### 2.5. Detection of Actual Samples

To investigate the practicability of the developed HCOOH sensor, the sensor was applied to detect HCOOH in triethylamine solution, which is usually used as the reaction solution in CO_2_ chemocatalysis and electrocatalysis systems. Three parallel samples were obtained and underwent the standard recovery test. The results are shown in Table 2. The recovery of HCOOH ranged from 94.77% to 107.73%, and the RSD was from 0.2% to 4.9%, demonstrating that the fluorescence sensor possesses practicability with promise for future applications.

## 3. Materials and Methods

### 3.1. Materials and Apparatus

HCOOH, HCOONa, HCOOEt_3_N, methanol, ethanol, tert-butanol, nicotinamide adenine dinucleotide disodium salt (NADH), oxalic acid, sodium sulfate, glucose, KCl, acetic acid HAc, sodium acetate NaAc, NaH_2_PO_4_, Na_2_HPO_4_, sodium citrate, and NaCl were purchased from the Shanghai Aladdin Biochemical Technology Co. (Shanghai, China). 2-Bromopropionic acid, 4-bromobutyric acid, mandelic acid, and aminocaproic acid NH_2_(CH_2_)_5_COOH were acquired from the MACKLIN reagent (Shanghai, China). The Ir-complex PP-Ir-Cl and the fluorescence probe P-coumarin were synthesized according to our previous work [23,24]. The UHQ II system (Elga) was used to purify water to a resistivity of 18 MΩ·cm for the preparation of all solutions. Phosphate buffer solution (PBS) was prepared using Na_2_HPO_4_ and NaH_2_PO_4_. HAc-NaAc buffer solution was prepared using HAc and NaAc. Fluorescence spectra were collected using an F-4700 fluorescence spectrophotometer (HITACHI, Tokyo, Japan).

### 3.2. Detection Protocol and Spectrophotometric Analysis

In a typical detection experiment, 990 µL of 0.2 M phosphate buffer with different pH or an HAc-NaAc buffer containing 10 *v*/*v*% of different organic solvents, 100 uM of the fluorescence probe P-coumarin, and different amounts of the Ir-complex PP-Ir-Cl was added to different concentrations of HCOOH or formate solution and incubated for 1 h at 60 °C. The fluorescence intensity of each sample was then tested. Finally, a standard curve line was constructed between various concentrations of HCOOH or formates and the recovery values of fluorescence intensity. In addition, the fluorescence probe selectivity was studied under the optimal conditions using various interfering substances instead of HCOOH or formates.

### 3.3. Detection of Actual Samples

The fluorescence sensor had excellent specificity for HCOO^−^. To study the performance of the sensor in a CO_2_ hydrogenation reaction, the sensor was applied to detect the produced HCOOH in a triethylamine environment. Briefly, 1.0 mL of triethylamine solution containing various concentrations of HCOOH (50 µM, 100 µM, 150 µM, 200 µM, and 300 µM) was first removed of free triethylamine using a vacuum distillation, and then 1.0 mL of HAc-NaAc buffer (0.2 M, pH = 4.0) containing 10 *v*/*v*% of methanol, 100 uM of the fluorescence probe P-coumarin, and 5.0 uM of the Ir-complex PP-Ir-Cl was added and incubated for 1.0 h at 60 °C. Finally, the fluorescence spectroscopy of each sample was tested. Three experiments were performed in parallel, and the RSD was calculated.

## 4. Conclusions

In conclusion, we have developed a facile and rapid fluorescence sensor for HCOOH based on highly active Ir-complexes’ catalytic transfer hydrogen fluorescence turn-on reaction and using HCOOH or formates as the hydrogen source. The excellent HCOOH-assisted catalytic fluorescence response system was obtained by optimizing various experimental conditions, including pH, reaction solvents, temperature, catalyst amount, and reaction time. The proposed HCOOH sensor shows good sensitivity and specificity for the detection of HCOOH, HCOONa, and HCOOEt_3_N, with a satisfactory linear range and a low LOD (50.6 ppb), and it can achieve an accurate determination of HCOOH samples in a CO_2_ catalytic hydrogenation system. Therefore, the HCOOH sensor could be used as a new detection method for HCOOH and can provide a new design path for other sensors.

## Data Availability

Not applicable.

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
