# Peer review of "Sensitive and Facile HCOOH Fluorescence Sensor Based on Highly Active Ir Complexes’ Catalytic Transfer Hydrogen Reaction"

_molecules, 2022, doi:10.3390/molecules27217431_

Round 1

Reviewer 1 Report

This work designed a sensitive and facile HCOOH sensor based high active catalytic hydrogeneration organometallic catalysts and quinone reduction fluorescence “turn-on” probe. In the presence of HCOOH or formate, organometallic catalysts could quickly catalyze the probe to produce strong fluorescence signal. In additions, this sensor presented good detection accuracy, excellent sensitivity and specificity for HCOOH and formats comparing with most previous methods.

In my opinion, this work has provided a new strategy to build HCOOH sensors using high active Ir-complexes catalytic hydrogeneration reaction, which is interested and applied for HCOOH detection. Thus, I would like recommend this paper to be published after minor revision.

The following points should be addressed:

When the authors use an abbreviation, its meaning should be clarified during the first time (e.g. "HPLC", "NMR"), and the abbreviations should be used in the following paper. All the terminological expressions should be uniform in the whole manuscript. Please carefully check and correct.

Check that the substance name format is correct, e.g. (NAD+).

English should be improved. e.g. "The hydrogen sources (such as HCOOH, H2) was also is also essential besides cata-lysts and substrates", should be modified to “​The hydrogen sources (such as HCOOH, H2) were also essential besides catalysts and substrates", "formats" is misspelled.

In Figure 4, it is better to indicate the concentration of HCOOH and other interfering molecules.

The data format in Table 2 should be uniform. e.g. "49.962" should be modified to "49.96".

The reproducibility of sensor preparation should be inserted in the manuscript.

Figure 3 lack error bars.

Author Response

Point-by-point responses to reviewers’ comments.

Response to referee 1:

Comments and Suggestions for Authors: This work designed a sensitive and facile HCOOH sensor based high active catalytic hydrogeneration organometallic catalysts and quinone reduction fluorescence “turn-on” probe. In the presence of HCOOH or formate, organometallic catalysts could quickly catalyze the probe to produce strong fluorescence signal. In additions, this sensor presented good detection accuracy, excellent sensitivity and specificity for HCOOH and formats comparing with most previous methods.

In my opinion, this work has provided a new strategy to build HCOOH sensors using high active Ir-complexes catalytic hydrogeneration reaction, which is interested and applied for HCOOH detection. Thus, I would like recommend this paper to be published after minor revision.

 Response: Thank you very much for your professional suggestions and reminds on our work. Following your suggestions, we revised our manuscript carefully.

Question 1: When the authors use an abbreviation, its meaning should be clarified during the first time (e.g. "HPLC", "NMR"), and the abbreviations should be used in the following paper. All the terminological expressions should be uniform in the whole manuscript. Please carefully check and correct.

Answer 1: Thank you for your professional suggestions. According to your suggestions, we have made the changes in line 36-37 of the manuscript and checked all abbreviations to make sure they have meaning before we use them.

Question 2: Check that the substance name format is correct, e.g. (NAD+).

Answer 2: Thank you for your professional suggestions. According to your suggestions, we have corrected the NAD+ in line 59 of the manuscript.

Question 3:  English should be improved. e.g. "The hydrogen sources (such as HCOOH, H2) was also is also essential besides catalysts and substrates", should be modified to “The hydrogen sources (such as HCOOH, H2) were also essential besides catalysts and substrates", "formats" is misspelled.

Answer 3: Thank you for your careful reminds. According to your suggestions, we have corrected the grammar in line 68 of the manuscript.

Question 4: In Figure 4, it is better to indicate the concentration of HCOOH and other interfering molecules.

Answer 4: Thank you for your professional suggestions. We have indicated the concentrations of the various substances in Figure 4. The concentration of the HCOOH and HCOONa was 1.0 mM, the other interfering substances was 5.0 mM.

Question 5: The data format in Table 2 should be uniform. e.g. "49.962" should be modified to "49.96".

Answer 5: Thank you for your careful reminds. According to your suggestions, we have made the changes in Table 2.

Table 2. Recoveries for detecting HCOOH in real samples (n = 3).

Sample

Added(μM)

Found(μM)

Recovery(%)

RDS(%)

1

50

49.96

99.92

0.4

2

100

94.77

94.77

0.2

3

150

149.95

99.97

4.9

4

200

215.45

107.73

2.7

5

300

311.14

103.72

4.0

Question 6: The reproducibility of sensor preparation should be inserted in the manuscript.

Answer 6: Thank you very much for your important reminds. According to your suggestions, we have supplemented this with reproducible experiments and corresponding description in the revised manuscript (page 7, lines 251-255).

Figure 4. A) The interference of various organic acid and ions for he developed HCOOH fluorescence sensor. The concentration of the HCOOH and HCOONa was 1.0 mM, the other interfering substances was 5.0 mM. B) Reproducibility of sensor response to HCOOH fluorescence. Experiment conditions:1.0 mM HCOOH, 0.1 M PBS solution containing 10 v/v% MeOH, pH = 4.0, 5.0μM Ir-complex, reaction time 30 mins, reaction temperature 60 ℃.

Question 7: Figure 3 lack error bars.

Answer 7: Thank you very much for your important reminds. Figure 3 shows the error bar added. The error bar cannot be shown because the error of the three parallel experiments is very small.

Reviewer 2 Report

In the manuscript, the authors reported a fluorescent sensor for HCOOH based on Ir-complexes catalytic transfer hydrogen fluorescence turn-on reaction using HCOOH or formate as hydrogen sources. The sensor also has been applied for detecting HCOOH in triethylamine solution. However, I think the manuscript can not be accepted due to the following issues.

1.      The selectivity of the sensor to HCOOH is the major concern. Why can't the sensor system respond to other organic acids? Moreover, according to the selectivity studies, the species with reduction ability could also induce the fluorescence response.

2.      The sensing process was performed in acid condition (pH =4.0). Under alkaline conditions, the fluorescence of sensor could be turned on by due to the ester group could be hydrolyzed. However, in the practical application, the sensor has been applied for selectively detecting HCOOH in triethylamine solution (alkaline conditions).

3.      There are many typing mistakes. For example, line 263, “4. results” should be changed to “4. Conclusion”; line 254 “3.5. Detection of N2H4 in CO2 hydrogeneration reaction solution”, no “N2H4” has been mentioned in the application; line 183, “and formats” should be changed to and formates”.

Author Response

Response to referee 2:

Comments and Suggestions for Authors: In the manuscript, the authors reported a fluorescent sensor for HCOOH based on Ir-complexes catalytic transfer hydrogen fluorescence turn-on reaction using HCOOH or formate as hydrogen sources. The sensor also has been applied for detecting HCOOH in triethylamine solution. However, I think the manuscript can not be accepted due to the following issues. Response: Thank you very much for your professional suggestions and reminds on our work. According to your suggestions, we have carefully revised the manuscript and added some experiments. We hope that the revised manuscript could get your approval and meet the requirements of this article published on Molecules.

Question 1: The selectivity of the sensor to HCOOH is the major concern. Why can't the sensor system respond to other organic acids? Moreover, according to the selectivity studies, the species with reduction ability could also induce the fluorescence response.

Answer 1: Thank you very much for your important questions. Firstly, the developed HCOOH sensor was based high-efficient Ir-complexes catalytic hydrogeneration reaction, and a large number of researches have demonstrated HCOOH or its anion was an irreplaceable source of hydrogen in the process of catalytic hydrogen transfer except for biological coenzymes nicotinamide adenine dinucleotide (NADH) according to the corresponding catalytic mechanism (Catal. Sci. Technol., 2021, 11, (24), 7982-7991; Green Chem., 2018,20, 2118-2124; Green Chem., 2013, 15, 629-634; ACS Catal., 2016, 6, 2637-2641.). In additions, other organic acids have also proved to be not used as a hydrogen source of Ir-complexes catalytic hydrogeneration reaction in our experiments (Figure 4A). Figure R1. A proposed mechanism of Ir-complexes catalytic hydrogeneration for ketones.

Figure 4. A) The interference of various organic acid and ions for he developed HCOOH fluorescence sensor. The concentration of the HCOOH and HCOONa was 1.0 mM, the other interfering substances was 5.0 mM. B) Reproducibility of sensor response to HCOOH fluorescence. Experiment conditions:1.0 mM HCOOH, 0.1 M PBS solution containing 10 v/v% MeOH, pH = 4.0, 5.0 μM Ir-complex, reaction time 30 mins, reaction temperature 60 ℃.

According to the selectivity studies, the species with reduction ability could also induce the fluorescence response. We agree with some strong reduction ability species such as sodium borohydride NaBH4, NADH could reduce quinone to hydroquinone and lead to the fluorescence turn-on of probe. This worry was also a common difficulty for redox sensor (Trends in Analytical Chemistry., 2020, 133, 116112.). Thus, the strong reducing substances should be avoided when the developed HCOOH sensor was used to detection HCOOH and formats. 

Question 2: The sensing process was performed in acid condition (pH =4.0). Under alkaline conditions, the fluorescence of sensor could be turned on by due to the ester group could be hydrolyzed. However, in the practical application, the sensor has been applied for selectively detecting HCOOH in triethylamine solution (alkaline conditions).

Answer 2: Thank you very much for your important questions. For the developed HCOOH sensor, acid condition (pH =4.0) has been proved to be optimal pH due to signal to noise ratio. Thus, for the alkaline actual sample detection, pretreatment of actual sample (remove excess triethylamine solvent by vacuum distillation or adjusting the pH to 4.0 with HCl) was necessary. In this manuscript, the excess alkaline triethylamine was all removed by vacuum distillation for each actual sample detection. Specific experiment methods have also been described in the Materials and Methods section of manuscript (page 3-4, lines 113-121.).

Question 3: There are many typing mistakes. For example, line 263, “4. results” should be changed to “4. Conclusion”; line 254 “3.5. Detection of N2H4 in CO2 hydrogeneration reaction solution”, no “N2H4” has been mentioned in the application; line 183, “and formats” should be changed to “and formates”.

Answer 3: Thank you for your professional suggestions. We have carefully checked and revised the manuscript according to your suggestions.  The "4. Results " have be changed to " 4. Conclusion " and " formats " be changed to" formates ". In additions, we have also changed " Detection of N2H4 in CO2 hydrogeneration reaction solution " to " Detection of actual samples".

Round 2

Reviewer 2 Report

After the revisions, the manuscript can be accepted.